# Development of a Versatile, Near Full Genome Amplification and Sequencing Approach for a Broad Variety of HIV-1 Group M Variants

**DOI:** 10.3390/v11040317

**Published:** 2019-04-01

**Authors:** Andrew N. Banin, Michael Tuen, Jude S. Bimela, Marcel Tongo, Paul Zappile, Alireza Khodadadi-Jamayran, Aubin J. Nanfack, Josephine Meli, Xiaohong Wang, Dora Mbanya, Jeanne Ngogang, Adriana Heguy, Phillipe N. Nyambi, Charles Fokunang, Ralf Duerr

**Affiliations:** 1Department of Pathology, New York University School of Medicine, New York, NY 10016, USA; andybanin@gmail.com (A.N.B.); Michael.Tuen@nyumc.org (M.T.); bimelajude@gmail.com (J.S.B.); Paul.Zappile@nyumc.org (P.Z.); a_nanfack@yahoo.fr (A.J.N.); Adriana.Heguy@nyumc.org (A.H.); camacf01@nyumc.org (P.N.N.); 2Faculty of Medicine and Biomedical Sciences, Department of Biochemistry, University of Yaoundé 1, Yaoundé BP 1364, Cameroon; jngogang@yahoo.fr; 3Faculty of Science, Department of Biochemistry, Yaoundé BP 1364, Cameroon; 4Center of Research for Emerging and Re-Emerging Diseases (CREMER), Institute of Medical Research and Study of Medicinal Plants, Yaoundé BP 906, Cameroon; marcel.tongo@gmail.com; 5Applied Bioinformatics Laboratories (ABL) and Genome Technology Center (GTC), Division of Advanced Research Technologies (DART), New York University Langone Medical Center, New York, NY 10016, USA; alireza.khodadadi-jamayran@nyumc.org; 6Medical Diagnostic Center, Yaoundé BP 15810, Cameroon; jmeli_cm@yahoo.fr; 7Chantal Biya International Reference Center for Research on HIV/AIDS Prevention and Management, Messa Yaoundé BP 3077, Cameroon; 8Manhattan Veterans Affairs New York Harbor Healthcare System, New York, NY 10010, USA; Xiaohong.Wang@va.gov; 9Faculty of Medicine and Biomedical Sciences, Department of Microbiology, Parasitology and Infectious Diseases, University of Yaoundé 1, Yaoundé BP 1364, Cameroon; dmbanya1@yahoo.co.uk; 10Faculty of Medicine and Biomedical Sciences, Department of Pharmacotoxicology & Pharmacokinetics, University of Yaoundé 1, Yaoundé BP 1364, Cameroon; charlesfokunang@yahoo.co.uk

**Keywords:** Near full genome amplification and sequencing, HIV-1 group M subtype-independent approach, rational primer design, single-genome amplification (SGA), third-generation sequencing (TGS), bulk sequencing and cloning

## Abstract

Near full genome sequencing (NFGS) of HIV-1 is required to assess the genetic composition of HIV-1 strains comprehensively. Population-wide, it enables a determination of the heterogeneity of HIV-1 and the emergence of novel/recombinant strains, while for each individual it constitutes a diagnostic instrument to assist targeted therapeutic measures against viral components. There is still a lack of robust and adaptable techniques for efficient NFGS from miscellaneous HIV-1 subtypes. Using rational primer design, a broad primer set was developed for the amplification and sequencing of diverse HIV-1 group M variants from plasma. Using pure subtypes as well as diverse, unique recombinant forms (URF), variable amplicon approaches were developed for NFGS comprising all functional genes. Twenty-three different genomes composed of subtypes A (A1), B, F (F2), G, CRF01_AE, CRF02_AG, and CRF22_01A1 were successfully determined. The NFGS approach was robust irrespective of viral loads (≥306 copies/mL) and amplification method. Third-generation sequencing (TGS), single genome amplification (SGA), cloning, and bulk sequencing yielded similar outcomes concerning subtype composition and recombinant breakpoint patterns. The introduction of a simple and versatile near full genome amplification, sequencing, and cloning method enables broad application in phylogenetic studies of diverse HIV-1 subtypes and can contribute to personalized HIV therapy and diagnosis.

## 1. Introduction

HIV-1 full genome sequencing is a challenging task due to the broad degree of HIV-1 genomic diversity worldwide. Four major groups of HIV-1 exist, groups M, N, O, and P [1,2,3], which are highly divergent and trace back to separate cross-species transmission events from primates to humans [4]. HIV-1 group M has spread globally, causing more than 85% of global HIV infections and can be subdivided into nine subtypes (A–D, F–H, J, and K), at least six sub-subtypes of A and F (A1–A6, F1–F2), currently at least 98 circulating recombinant forms (CRFs) [5], and numerous unique recombinant forms (URFs) [6]. CRFs and URFs are composed of two or more (sub)-subtypes; while CRFs have been identified in at least three epidemiologically unlinked individuals, URFs are defined as recombinants without evidence of onward transmission [5]. The continuous emergence of novel HIV-1 M strains [6,7,8,9] including sequences that remain unclassified or can be attributed to ancient strains [10] poses a significant challenge for HIV-1 molecular surveillance, diagnosis, and therapy [11,12]. Consequently, robust and versatile amplification and sequencing protocols are needed that apply to such strains over the near full genome (NFG).

The vast majority of sequences in the Los Alamos National Library (LANL) HIV sequence database are derived from partial genome sequencing [13]. Partial genome sequencing obscures subtype determinations and limits the identification of epidemiologic signatures and recombination breakpoints, which can be distributed across the entire viral genome [14,15]. To overcome these limitations, PCR-based near full genome sequencing (NFGS) protocols have been introduced, which were mostly applied in cohorts restricted to one or a few subtypes [16,17,18,19,20]. Protocols have been optimized for next-generation sequencing (NGS) [21,22], or the use of oligonucleotide probes combined with real-time PCR [23,24]. The coverage of multiple subtypes has been addressed by universal NFGS approaches [22,25]. Universal approaches rely on primer binding to semi-conserved regions in the HIV genome, and limited coverage and efficiencies can still occur depending on viral load and subtype. The combination of NGS with HIV-specific probes increased detection sensitivity and sequencing depth [26]; however, the detection of emerging and rapidly evolving strains is expected to be challenging. NGS approaches provide large-scale sequencing with in-depth coverage and analyses of minority variants; however, their usage is limited in resource-constrained settings due to the required equipment and expertise, and do not allow downstream processing of the amplicons and cloning. There is still a need for robust and straightforward NFGS protocols suited for a broad diversity of HIV-1 M subtypes and the generation of NFG amplicons adaptable to downstream biological applications.

A flexible NFGS approach has therefore been established by our team using rationally designed primers that bind at highly conserved domains across major HIV-1 group M subtypes. The set of primers enables nested PCRs to span single or multiple functional gene regions. It provides NFG amplification and sequencing through one-amplicon, two-amplicon or multiple-amplicon approaches. Pure HIV-1 M subtypes and highly diverse HIV-1 M URFs from Cameroon, characterized in detail in Banin et al. [27], were used to validate the protocol, which was applied to bulk sequencing, cloning, single-genome amplification (SGA), and third-generation sequencing (TGS).

## 2. Materials and Methods

### 2.1. Ethical Clearance

This study was conducted in accordance with the Declaration of Helsinki and the protocol was approved by the Institutional Ethical Review Board of the Cameroon Ministry of Public Health and by the Institutional Review Board at New York University School of Medicine (NYUSoM), New York, USA on April 6, 2018 (protocol: i09-0431). Before sample collection, informed consent was obtained from the study participants, who were all part of a cohort of HIV positive individuals; this cohort is monitored at the Medical Diagnostic Center (MDC) in Yaoundé, Cameroon in collaboration with the New York University School of Medicine (NYUSoM), New York, USA.

### 2.2. Study Samples

Whole blood samples were collected at the MDC between 2000 and 2015. They were shipped under standard regulatory conditions for biological materials to NYUSoM for processing of plasma and peripheral blood mononuclear cells (PBMCs) using Ficoll gradient centrifugation (Histopaque, Sigma-Aldrich, St. Louis, MO, USA), and stored at −80 °C. Study samples were selected based on previously identified URF sequences through partial genome sequencing in our group and on the availability of plasma volumes ≥500 µL [28,29,30].

### 2.3. RNA Extraction

Viral RNA was extracted from 500 µL HIV-1 positive plasma using QIAamp Viral RNA Minikit (Qiagen, Valencia, CA, USA). Virions were concentrated by ultracentrifugation at 14,000× *g* for 2 h at 4 °C. All but 140 µL of the supernatant was removed before addition of lysis buffer and subsequent RNA extraction according to the manufacturer’s instructions. Extracted RNA (30 µL) was either immediately reverse transcribed or stored at −80 °C.

### 2.4. cDNA Synthesis

Reverse transcription of RNA was performed using SuperScript III (Life Technologies, Carlsbad, CA, USA) with primer OFM19 [31] corresponding to HXB2 position 9632–9604. If amplicons within the 5’ half of the HIV-1 genome could not be obtained using cDNA generated with OFM19 primer, HIV 6352 rev [25] was used as an alternative. For the generation of cDNA, 5.25 µL nuclease-free water (Promega) was mixed with 0.75 µL reverse primer OFM19 or HIV 6352 rev (20µM), 3 µL (10 mM) dNTP mix and 30 µL extracted template RNA. The mix was incubated at 65 °C for 5 min and removed on ice for 1 min. 12 µL 5× first strand buffer, 3 µL (100 mM) DTT solution, 3 µL (40 U/µL) RNaseOUT, and 3 µL (200 U/μL) SuperScript III Reverse Transcriptase were added to the mix to get a total volume of 60 µL. The mix was vortexed and incubated at 50 °C for 1 h followed by an increase to 55 °C for 1 h. SuperScript III Reverse Transcriptase was inactivated at 70 °C for 15 min. Remaining RNA was digested by adding 3 µL RNase H (2 U/µL) and incubation at 37 °C for 20 min.

### 2.5. Primer Design

The rational design of universal HIV-1 group M amplification and sequencing primers was based on full genome multiple sequence alignments with reference strains downloaded from the LANL database. Our approach is related to PrimerDesign-M, an alignment-based primer design tool from the LANL database, which also served as the first guide to identify appropriate genomic regions for some of our primers. Generally, most primers were designed manually based on sequence alignments and primer criteria as follows. An HIV-1 full genome alignment comprised of 480 reference sequences from all major group M (sub)-subtypes and CRFs [32] was shortened to a representative 40 strain-panel. The latter included four references per eight (sub)-subtypes A1, A2, B, C, D, F1, F2, G, and two CRFs, 01_AE, and 02_AG (Results *3.2*), which make up to the most dominant (sub-)subtypes/CRFs prevailing the five inhabited continents of the world.

The procedure was similar for the design of amplification and sequencing primers with the only difference of targeted regions. Amplification primers mostly targeted conserved domains adjacent or at the ends of the genes of interest. Sequencing primers had no spatial constraints; however, appropriate distances (~500 bp) between consecutive primer binding sites should provide sufficient sequence overlap between the multiple complementary sequence fragments (≥100 bp overlap) while aiming for the lowest possible number of primers to cover NFGS. Preferentially, primers were designed to have a length of 18–30 base pairs with 40–60% G/C content and a melting temperature (Tm) of 50–65 °C. Primer pairs should be within 5 °C difference in Tm and exempt of complementary regions to avoid heterodimers. Primers with long stretches of identical base pairs (especially G and C), hairpin structures, or self-complementary regions (homodimers) were avoided, as can be analyzed with a primer design software such as IDT OligoAnalyzer (https://www.idtdna.com/calc/analyzer). If possible, primers should start and end with one or two G/C base pairs. The 3’ end of the primer was most critical to accurately match the homologous regions of the majority of reference sequences. Few ambiguous base pairs were tolerated in the central region of the primer sequence and up to three ambiguity characters were introduced for ≤70% conserved sites per primer. Since accurate primer design and positive prediction do not guarantee efficient, practical implementation, trial and error have to succeed. Amplification primers were screened with at least three study samples (cDNA templates), and if possible, in conjunction with already established pairing primers. Primer selection was based upon efficient PCR amplification yielding strong and clean single bands without smear or evidence of unspecific amplification products in at least one NFG PCR or at least 1/3 of PCR reactions amplifying smaller regions. Out of >70 newly tested amplification primers, 26 primers passed the selection criteria (Results *3.2*). Sequencing primers were screened with at least two study samples, i.e., bulk-amplified nested PCR products with single strong bands that enabled positive sequencing reactions using established sequencing primers. Out of >90 newly tested sequencing primers, 64 primers worked sufficiently to determine sequences ≥700 bp in length (quality value ≥16, Macrogen sequencing service, New York, NY, USA) from half of the test samples (Results *3.2*) [33].

### 2.6. PCR Amplification

PCR reactions were performed with PrimeSTAR GXL DNA polymerase, (Clontech, Mountain View, CA, USA) or High-Fidelity Taq polymerase (Life Technologies, Carlsbad, CA, USA). For Prime polymerase, nested PCRs were performed in a total reaction volume of 25 µL per sample containing 15.5 µL nuclease-free water, 5 µL 5X PrimeSTAR GXL Buffer (Mg^2+^ plus), 2.5 µL dNTP mix (2.5 mM each), 0.5 µL Prime polymerase, 0.5 µL forward and reverse primers, and 1 µL cDNA. For the Taq polymerase, a total reaction volume of 25 µL per sample constituted of 18.85 µL nuclease-free water, 2.5 µL buffer 10X, 1 µL MgSO_4_, 0.5 µL dNTPs, 0.15 µL Taq polymerase, 0.5 µL forward and reverse primers, and 1 µL cDNA.

The PCR reactions were performed with the following cycling conditions: initial denaturation step of 30 s at 98 °C and 30 cycles of 10 s at 98 °C (denaturation), 15 s at 53 °C (annealing), and elongation at 68 °C for 1 min per 1 kb of the amplification product. After 30 cycles, a final elongation step was performed at 68 °C, and the mix stored at 4 °C. The PCR products were analyzed by gel electrophoresis at 120 V for 30 min on a 0.8% agarose gel. To exclude/minimize unspecific amplification and cross-contamination, samples from different participants were processed separately or sequentially, and nucleotide removal treatments were applied frequently. Multiple sequence alignments (MEGA5.2) and highlighter plots (Los Alamos HIV Database) were done with current and historical PCR products from our lab to exclude cross-contamination.

### 2.7. Sanger Sequencing and Sequence Editing

Sanger Sequencing of amplified PCR products was done by Macrogen (Macrogen Corp., New York, NY, USA). Assembly and editing of the sequences were done with DNAStar package programs SeqMan Pro and EditSeq (Lasergene, Madison, WI, USA).

### 2.8. Simplot and Recombinant Drawing Tool

The Simplot software package version 3.2 (Window size 200, step size 20 and 250 bootstrap replicate) was used for breakpoint determination. Schematic illustrations of the various full genomes were done using the recombinant drawing tool provided by the Los Alamos HIV sequence database. A bootstrap support of 70% between a subtype reference strain and a query sequence was used as criteria to assign a subtype to a breakpoint region. If bootstrap values remained below 70% using several pure and CRF reference subtypes, the region was considered unidentified.

### 2.9. Third-Generation Sequencing

Third-generation sequencing (TGS) was performed using Oxford Nanopore MinIon flow cells, capable of high-throughput sequencing in real-time and generating long DNA reads. TGS was carried out at the NYU sequencing and genomic unit. Unsheared HIV DNA from 12 samples was quantitated with the qubit HS DNA kit (Thermo Fisher Scientific). DNA was normalized to 1 µg in 45 uL of buffer and individually input into the Oxford Nanopore PCR Barcoding kit (EXP-PBC001, R9 Version) according to the protocol outlined by Oxford Nanopore (PBGE_9007_V7_revA_16May2016\ PBGE_9007_v7_revJ_16May2016). Each sample was barcoded with a unique adapter index provided in the kit (using the NEB LongAmp Taq master mix, NEB # M0287S) and run with the following PCR conditions: Initial Denaturation for 3 min at 95 °C, Denaturation for 15 s at 95 °C, Annealing for 15 s at 62 °C, Extension for 12 min at 65 °C, Denaturation to Extension was repeated for an additional 14 cycles to compensate for the large fragment sizes (15 cycles total), and one final extension at the end for 14 min at 65 °C. After PCR, each sample was cleaned with Ampure XP (Beckman Coulter) and barcoded samples were quantitated with the Qubit HS DNA kit. Twelve samples were normalized to 110 ng each and combined into one pool for input into the Nanopore Sequencing kit from Oxford Nanopore (SQK-NSK007). The library prep was followed, as described in the protocol above. The pooled library was eluted for 30 min at 37 °C. The final library was quantitated with the qubit HS DNA kit and added onto a primed SpotON Flow Cell (Oxford Nanopore, #FLO-MIN105). The SpotON Flow cell was run on the MinION Mk 1 (#MIN-MAP002) with the script NC_48Hr_Sequencing_Run_FLO_MIN105. FASTQ files were generated using poretools (version 0.6.0) (PMID: 25143291) and general quality controls were performed. Barcodes were trimmed using cutadapt (version 1.9.1) (http://journal.embnet.org/index.php/embnetjournal/article/view/200) and all the reads were mapped to HIV-I (HXB2) reference genome (K03455.1) for an initial quality assessment using the Burrows-Wheeler Aligner (BWA) (version 0.7.13) mem algorithm [34] and visualized using Integrative Genomics Viewer (IGV) software packages [35]. Phylogenetic analyses were done using MEGA5.2 software [36]. Maximum likelihood phylogenetic trees were generated using 1000 bootstrap replicates [37]. Any HIV unspecific reads or minor HIV variants that clustered with a viral population from an unrelated sample were removed. TGS (sub)populations were averaged to consensus (con) sequences using Consensus Maker (Los Alamos National Library (LANL) Database) (www.hiv.lanl.gov) or SeqMan Pro.

### 2.10. Single-Gnome Amplification

Single genome amplifications (SGA) are PCRs performed on endpoint-diluted cDNA to preclude amplification artifacts such as template switching that can occur when mixtures of genetically diverse templates are used. Template cDNA dilutions that resulted in no more than 30% positive nested PCR reactions were assumed to contain one amplifiable cDNA template per positive PCR (>80% likelihood) [31]. Specifically, ten nested PCR reactions were performed per cDNA dilution. Multiple cDNA dilutions in the range between 1:2 and 1:1,000,000 were tested for each sample. Dilutions with no more than three positive PCR reactions were considered for SGA analyses and the positive amplicons subjected to direct sequencing. SGAs were performed using High-Fidelity PrimeSTAR GXL DNA polymerase (Clontech, Mountain View, CA, USA) using the same PCR and cycling conditions as described under *2.5*.

### 2.11. Cloning

PCR products were ligated into pcDNA3.1 (blunt-end cloning, PrimeSTAR PCR products) or pCR4 TOPO (3’-A overhang cloning, Taq PCR products) cloning vectors (Life Technologies, Carlsbad, CA, USA). The plasmids were subsequently transformed into One Shot Top10 Chemically Competent *E. coli*, (Life Technologies, Carlsbad, CA, USA) and cultured overnight using LB-ampicillin agar plates [38].

### 2.12. Data Storage and Documentation

Near full genome sequences are available from GenBank with accession numbers MK086109-MK086132. Whole sets of TGS sequences are available upon request. Correspondence, data and material requests should be addressed to Ralf Duerr (Ralf.Duerr@nyumc.org).

## 3. Results

### 3.1. Design of an Adaptable NFG Amplification Strategy

The goal was to establish a flexible and straightforward methodology for NFG amplification, cloning, and sequencing applicable to the broad variety of group M subtypes and recombinant forms. The protocol should be suitable for several downstream methods including high-throughput sequencing (NGS or TGS), SGA, cloning, and bulk sequencing, and applicable to modern practice in well-equipped labs as well as to basic procedures in resource-limited countries. The NFGS approaches aimed to cover >8.5 kb including all encoding HIV-1 M genes, i.e., from *gag* to *nef*. The primary goal was the reliable and consistent determination of NFGS of highly diverse group M strains irrespective of the number of amplicons needed (Figure 1). The first option was the amplification of HIV-1 NFG in a single PCR using primers targeting the 5’ UTR and 3’ UTR regions. Also, we established a two-amplicon and multiple-amplicon approach. The two-amplicon approach targeted two overlapping “half genome” (HG) sequences. The 5’ HG region (HG1) included *gag* and *pol*, while the 3’ HG region (HG2) included *env* and *nef*. The accessory genes (*vif*, *vpr*, *vpu*) were included in both half genomes in their overlapping part. The multiple amplicon strategies aimed to cover near full genomes using three to five overlapping amplicons from a selection of eight PCR constructs in total. Each PCR construct encompassed complete genomic units of encoding genes (*env* also separated into *gp120* and *gp41*).

### 3.2. Rational Primer Design for NFG Amplification and Sequencing of Diverse Group M Viruses

NFGS approaches on our highly diverse Cameroonian URF samples had only limited success using published NFG primers [16,25] (see below). Therefore, rational primer design was performed using full genome reference sequence alignments with diverse HIV-1 group M strains downloaded from the LANL database (Table 1). Initially, an alignment composed of 480 reference sequences was used belonging to several known pure and CRF clades covering the major branches of each subtype [32].

Subsequently, the alignment was reduced and simplified to a representative 40 strain-panel comprising four reference sequences per eight (sub)-subtypes A1, A2, B, C, D, F1, F2, G, and two CRFs, 01_AE, and 02_AG (Figure 2). The designed amplification primers mostly targeted conserved domains adjacent or at the ends of the HIV-1 M genes of interest under consideration of primer characteristics such as melting temperature, GC content and secondary structure (using IDT web application and/or PrimerDesign-M, LANL database). The newly designed primers were combined with published primers [25,31] that exhibited significant amplification efficiencies in our Cameroonian cohort (Table 1). The newly developed primers targeted regions in the HIV-1 M genome, which were highly conserved across several reference sequences of diverse subtypes, as shown for the NFG and two HG amplicon approaches (Figure 2; Appendix A). Of significant importance, the terminal nucleotides of the primer binding sites were highly conserved, whereas a few internal positions exhibited moderate variance for some primers/reference strains. The binding sites of three primers for half genome amplifications were fully conserved across the 40 reference strains panel (Appendix A). Overall, we designed/selected 26 new amplification primers, complemented by ten primers from previous studies [25,31] that proved to be efficient. They were used for the amplification of 15 different constructs including four NFGs, three HGs, and eight smaller regions within the full genome (Figure 1 and Figure 3, Table 1).

Sequencing primers were designed based on the same criteria as described above for amplification primers. They were designed to generate fragments with ≥100 bp overlap when sequences ≥700 bp are obtained (Appendix A). Seventy-one sequencing primers were applied including 43 newly designed forward primers and 21 newly designed reverse primers, complemented by seven reverse primers published by other groups [25,31] (Table 2 and Table 3).

Fourteen sequencing primers were most frequently used based on their broad reactivity with our diverse study samples (Appendix A). These 14 primers provided complete sequence coverage from HxB2 positions 596 to 9604 (~9.1 kb) in NFG one-amplicon approaches. For two-amplicon approaches, eight primers were used to sequence the 1st HG (HxB2 position 776 to 5956; ~5.2 kb) and six primers for the 2nd HG (HxB2 position 5037 to 9533; ~4.6 kb).

### 3.3. Subtype-Independent Amplification and Sequencing of HIV-1 Near Full Genomes

Initially, we screened published NFGS nested primers for suitability with our cohort samples from Cameroon. Nested PCRs with a primer set previously optimized for NFGS of clade C viruses [16] did not result in any NFG PCR products using three study samples including #6541-6, which worked for most other NFGS approaches tested in this study. However, using a nested primer set recently optimized for multiple HIV-1 subtypes [25], we obtained positive reactions for #6541-6 and two more study samples out of 23 tested specimens (Table 4). Based on these results, we screened the newly designed primers as well as combinations of our and published primers to optimize NFGS across multiple HIV-1 group M subtypes (Table 1).

To establish and optimize the method, we first used four study samples from infections with pure (sub)-subtype viruses (B and F2; #Strain0526 and #LB002-1), a CRF (CRF02_AG; #6506-8), and a URF (#6541-6) (based on preliminary and published partial genome sequencing data of Duerr and Nyambi labs [29,40]) (Figure 4). Subtypes F2 and CRF02_AG were chosen, because of their high prevalence in Cameroon [41]. Two of those constructs could be amplified with a one-amplicon approach, the other two with a two-amplicon approach (Table 4). Subsequently, we analyzed 19 more samples, including another subtype F2 virus (#LB022), another CRF02_AG virus (#LB006-1), and different highly diverse URF samples from Cameroon (Table 4). The 18 URF genomes were composed of subtypes A (A1), F (F2), G, and recombinants CRF01_AE, CRF02_AG, and CRF22_01A1, and revealed non-uniform mosaic NFG sequences [27].

Out of 23 samples, 11 NFGS were obtained with a one-amplicon approach. Another 11 samples needed two amplicons and one sample at least three amplicons for NFGS determination (Table 4). The one-amplicon approaches were based on nested PCRs with primers targeting the 5’ UTR and 3’ UTR ends to obtain approximately 9.1 kb products in a single PCR reaction (Figure 1). The four optimized NFGS primer pairs (NFG1 to NFG 4) obtained yields between 13% and 21.7 % (Table 1; Table 4); the best yield was obtained with the NFG1 approach based on four newly designed primers (21.7% yield). Only one sample (#6541-6) could be amplified using all four NFGS approaches (Table 4). The four one-amplicon approaches complemented each other to an overall 43% yield. In the two-amplicon approach, NFGS were obtained from two overlapping half genomes. Half genome 1 (HG1) consisted of *gag*, *pol* and the accessory genes *vif*, *vpr*, and *vpu* (~5.2 kb), whereas the second half genome (HG2) included *vif*, *vpr*, *vpu*, *env*, and *nef* (~4.6 kb) (Figure 1 and Table 1). HG1a and b had 45% and 80% yield, respectively, and complemented each other except for one sample, to achieve a 95% yield in combination. Of interest, HG2, which was obtained using a mix of newly designed primers and published primers [25,31] achieved 100% yield and proved to be the most efficient nested primer combination in our set of study samples (Table 4). The amplification of smaller regions for the multiple-amplicon approach included *gag* (1.5 kb), *gagpol* (4.5 kb), *pol* (3.2 kb), *vifvpu* (1.0 kb), *vif*/*gp120* (3.0 kb), *env* (3.4 kb), *gp120* (1,7 kb), and *gp41*/*nef* (1.8 kb) (Figure 1; Table 1 and Table 4) with yields >55%. Since plasma viral load (PVL) has been described as a critical parameter for successful NFG sequencing [16,17,21], we analyzed whether the viral load of a sample predicted the number of amplicons needed for characterizing the NFG. Using the diverse study set of Cameroonian URF viruses, no association was observed between the number of amplicons needed for NFGS and PVL (Appendix A). The lowest PVL in our set of study samples with determined NFGS was 306 cps/mL.

### 3.4. Comparative NFGS Analysis Using Different Amplification, Cloning, and Sequencing Strategies

To validate our NFGS approach, different amplification, cloning, and sequencing procedures including bulk amplification, high-throughput/deep sequencing, and SGA were applied. The NFGS outcome was compared across platforms using sample NYU6541-6, which gave positive results for most amplification approaches. Bulk sequencing is the traditional method for the genetic determination of viral variants that make up ≥20% of a viral population in a sample, *e.g*., it is still the standard for genotypic drug resistance testing [41,42]. High-throughput sequencing enables more in-depth analyses and can assess the diversity of viral populations in a sample. As proof of principle, we used the portable MinION TGS technology (Oxford Nanopore Technologies) with the capacity to determine long reads in real-time [43]. Despite limitations in accuracy and depth [44,45], it gave an estimate of whether our diverse amplicon approaches are suited for high-throughput/deep sequencing (Figure 5 and Table 5) [27].

Since amplification artifacts such as recombination events can occur *in vitro* during bulk amplification, single genome amplification (SGA) [31] was used for validation of our results. Six samples were tested for SGA including NFG and HG constructs. Our results confirmed the results reported by other groups [46,47] that major viral populations detected by deep or bulk sequencing show a good match with SGA sequences with comparable genetic coverage (Figure 5 and Table 5) [27]. Specifically, *in vitro* recombination rates <1.6% were reported for bulk amplification [46,47]. As a precaution, TGS minority variants with <2% prevalence that were not confirmed by SGA were consequently excluded in our analyses. In contrast to high-throughput sequencing such as NGS or TGS, bulk sequencing and SGA allow the cloning of NFG constructs, which enables further functional characterization in downstream biological assays, such as replicative fitness or phylogenetic drug resistance tests [48,49,50]. An NFGS plasmid generated by Topo cloning of an NFG bulk amplicon was included in the comparative analysis, which revealed comparable characteristics to bulk sequencing as evident by <0.1% genetic distance and similar recombination breakpoint patterns (Figure 5). A comparative Simplot Bootscan analysis was performed to underline the significance of the introduced NFGS approach for accurate subtype/recombinant form determination across platforms. Bulk sequencing, cloning, SGA, and TGS (consensus sequence) yielded similar NFGS results with genetic distances <2% across platforms and overall comparable recombinant breakpoints patterns (Figure 5).

## 4. Discussion

Here, we report a simple and efficient method for the amplification and sequencing of HIV-1 near full genomes from plasma to cover a broad range of recombinant (URF and CRF) viruses and pure group M (sub)-subtypes. Due to the flexible approach based on one, two, or more amplicons, 100% of the study samples were genotyped over the near full genome. Recent studies performing NFGS out of plasma achieved different efficiencies in the range of 45–100% [16,17,18,20,21,22,25,46], from which most approaches remained below 100% efficiency and/or were restricted to specific subtypes. A one-amplicon approach obtained 67% NFGS efficiency in a subtype C cohort [16], whereas a two-amplicon approach applied to samples of diverse clades yielded 90–92% success rates [25,46]. A study in a US clade B cohort used an alternative two- or three-amplicon strategy to obtain 76% yield for samples with viral loads >10,000 cps/mL. A four-amplicon strategy achieved 93% yield when applied to diverse HIV-1 subtypes and groups [21], however when used in an Indian cohort with clade C and AC recombinant samples, the yield dropped to 45% despite adjustments to an optimized six amplicon strategy [18]. The results of these studies imply and confirm our findings in achieving higher percentages of NFGS when diverse amplicon approaches are complementarily used. In our study, the 43% efficiency of the one-amplicon approach, based on four different NFG protocols, increased to 95% when a two-amplicon strategy based on two different half genome protocols was used, and eventually achieved 100% with the help of multiple amplicons. Two NGS-based methods were recently introduced, which also obtained 100% NFGS rates. They were applied in a clade B cohort using a four amplicon strategy [20] or in a highly diverse study population using mixed four- and six-amplicon NGS libraries [22]. The latter study used an elegant method based on degenerate primers fused to common adapter sequences for efficient library generation, which enabled broad coverage of HIV subtypes and groups; however, the need of NGS equipment and expert knowledge limits its application in resource-constrained countries. Also, the method required considerable optimizations with regards to RNA extraction and cDNA synthesis for suitability with clinical samples.

For NFGS, a viral load threshold of 3,000 cps/mL [16,21] or even 10,000 cps/mL [16,17,21] has been reported. NGS can generate full genome coverage at viral loads >log3.5 cps/mL, while even most sensitive NGS/probes approach gradually decline to partial genome coverage for <log3.5 cps/mL [22,26]. Of interest, we obtained NFGS over a broad range of PVL down to ~300 cps/mL, although it is worth mentioning that in our study, we only had one sample below 3,000 cps/mL PVL included, and plasma was routinely concentrated ~3.5 fold (from 500 µL to 140 µL) before RNA extraction. Other than that, our method needed no additional optimizations. Other groups reported that a substantially higher yield could be obtained using Benzonase pretreatment of plasma before RNA extraction to reduce the background of human RNA and DNA and by increasing the reverse transcription temperature from 42 °C to 47 °C [22]. Doubling the amount of RT enzyme from 200 U to 400 U significantly increased the critical RT step for NFGS in another study [17].

Despite the limitation in sample size (23 subjects), the analysis of a range of pure (sub)-subtypes and recombinant forms including A (A1), B, F (F2), G, CRF01_AE, CRF02_AG, and CRF22_01A1 implies a broad applicability of the presented NFGS approach across diverse (sub)-subtypes/recombinants and geographic regions of the world. Universal usage of our approach is further corroborated by the set of primers, designed to bind several most prevalent pure (sub)-subtypes and recombinant forms on the globe. Nevertheless, future studies are needed to show that the primers and the NFGS strategy are robust on larger sample sets from diverse regions of the world. The introduced procedure of rational primer design coupled with a flexible amplicon strategy allows researchers to improve the protocol further according to their needs.

Our approach provides a universal toolbox that can be used and adapted for different amplification and sequencing applications and is easy to handle for laboratories in resource-limited countries. Besides the flexible methodology, it allows both NFGS as well as focused molecular studies for distinct genomic regions. Since our method does not necessarily depend on high-throughput sequencing (NGS/TGS), which usually determines a one-way road towards genetic read-out, the SGA or bulk amplification can be used for the generation of molecular clones and subsequent downstream applications in biological assays. Notably, the introduced combination of primers from different studies [25,31] including our newly designed primers brought up highly efficient nested PCR primer pairs that may prove useful for purposes beyond NFGS and subtype monitoring. For example, we achieved 100% yield in our HG2 approach, which includes the amplification of the complete *env* region and accessory genes. It could be a useful alternative for *env* SGA, pseudovirus generation, screening for neutralizing antibody resistant strains, and genotypic tropism analysis. Another construct encompassing whole *pol*, *gag*, and accessory genes (HG1b) was obtained with 80% amplification yield when using four newly designed nested primers. The latter represents a promising approach for genotypic drug resistance testing in research and clinical care covering several essential target sites. Although not tested in the present study, we expect our NFG amplification approaches to performing well with state-of-the-art NGS systems to capture minority variants in quasispecies populations, as we showed recently for the sensitive Illumina MiSeq platform using a related amplification approach restricted to the reverse transcriptase region of *pol* [41].

Hence, the proposed protocols have broad applicability and can form the basis for downstream applications in phylogenetic and molecular studies, and further project-specific optimizations of primer combinations. While the studied in-depth sequencing approach based on MinION technology needs further optimization regarding accuracy and depth, it indicates future implications in real-time NFGS on-site.

## 5. Conclusions

In summary, we designed and validated a robust and adaptable protocol for NFGS from plasma, capable of amplifying and sequencing a broad diversity of HIV-1 group M strains including pure (sub)-subtypes and mosaic viral strains (URF and CRF). Despite the limited number of 23 samples, our approach suggests broad applicability based on the semi-conserved primer design and the flexible amplicon approach. Future studies using larger sample numbers from different geographic regions and epidemics will need to confirm its suitability in global HIV surveillance. The introduced NFGS approach may prove valuable to complement existing strategies in structural and functional HIV research including regions with rapid evolution and emergence of novel recombinant viral strains.

## Figures and Tables

**Figure 1 viruses-11-00317-f001:**
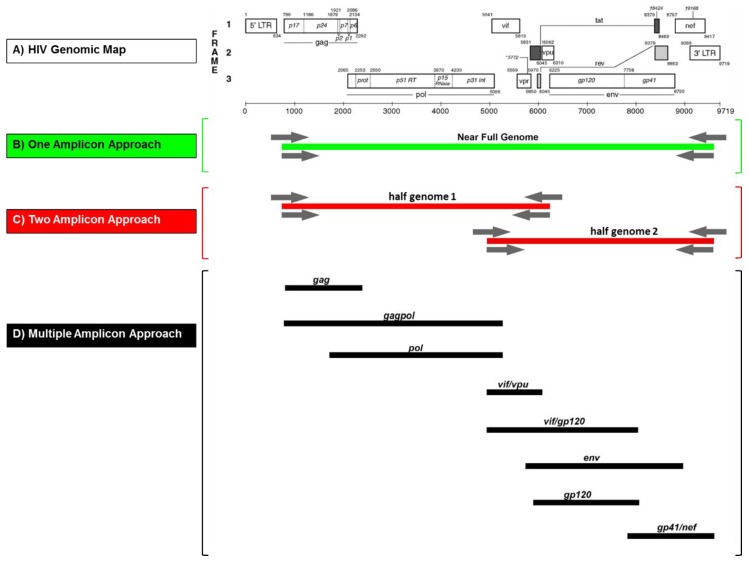
Near full genome sequence amplification approaches. (**A**) Genome map of HIV-1 reference strain HxB2 (GenBank: K03455). (**B**) One-amplicon approach to obtain a single 9.1 kb PCR fragment (near full genome) shown in green. (**C**) Two-amplicon approach with two overlapping PCR fragments (shown in red), with half genome 1 (HG1) 5.2 kb and half genome 2 (HG2) 4.6 kb in size. (**D**) Multiple-amplicon approach: Selection of eight overlapping PCR fragments (shown in black), i.e. *gag* (1.5 kb), *gagpol* (4.5 kb), *pol* (3.2 kb), *vifvpu* (1.5 kb), *vif*/*gp120* (3 kb), *env* (3.4 kb), *gp120* (1.7 kb) and *gp41*/*nef* (1.8 kb). Positions of primers are schematically shown with gray arrows in (A) and (B).

**Figure 2 viruses-11-00317-f002:**
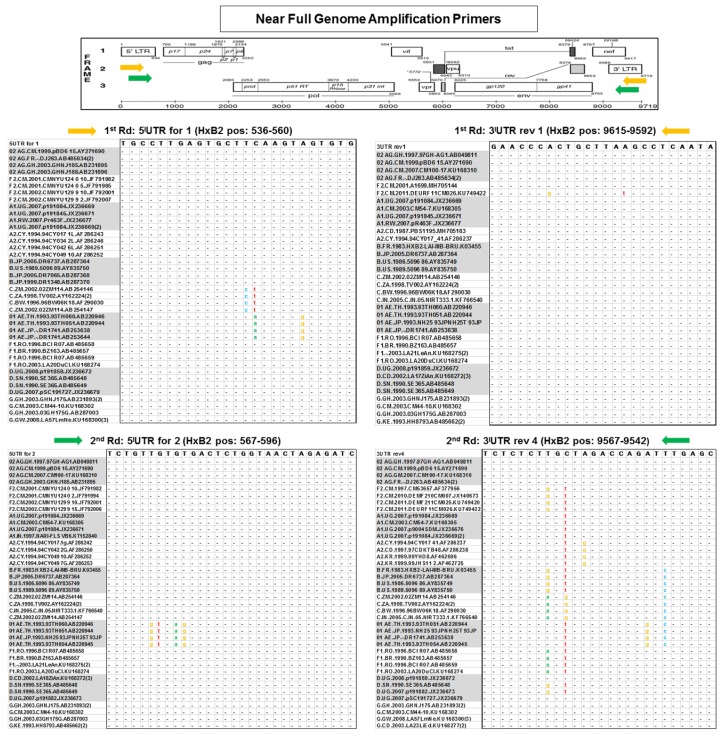
Conserved primer binding sites for HIV-1 near full genome amplification. Multiple sequence alignments of primer binding sites across ten different pure (sub)-subtypes and CRFs, as relevant for the near full genome amplification approach 1. For each (sub)-subtype/CRF, four reference sequences were selected that broadly cover the respective clade; for the binding site of primer 3’UTR rev 1, only two A2 and F2 reference sequences were available from the LANL database. The numbering of primer binding regions is based on the HxB2 reference strain (GenBank: K03455). Positions (pos) of primer binding sites are shown in brackets (HxB2 pos. 596–9542). Orange and green arrows along the HxB2 genome map indicate first round and second round primer positions, respectively. Black dots and colored letters indicate matches and mismatches with primer nucleotides shown on top, respectively.

**Figure 3 viruses-11-00317-f003:**
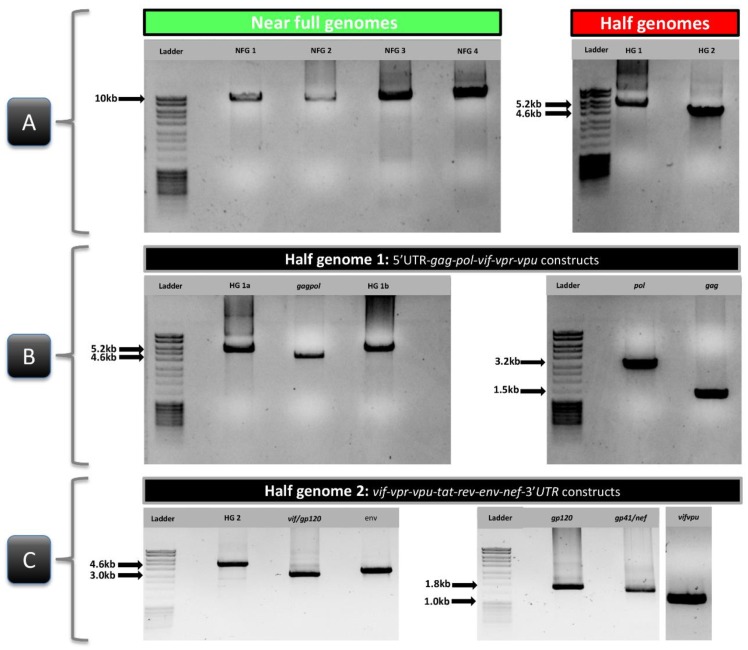
Gel electrophoresis of PCR products of different lengths for composite near full genome analysis. (**A**) HIV-1 near full genome (NFG; green) and half genome (HG; red) amplicons obtained with different primer sets. (**B**) Diverse amplicons within half genome 1 (HG1). (**C**) Diverse amplicons within half genome 2 (HG2). Amplicons were separated on an ethidium bromide-stained 0.8% agarose gel. MassRuler DNA ladder was used for amplicon size determination. The amplicons shown in the gels were obtained from the following samples: NYU6541-6 (NFG1), NYU129-5 (NFG2), LB089-1 (NFG3 and *env*), NYU1122-1 (NFG4, HG1, and HG1a), NYU119-3 (HG2, *pol*, and *gag*), NYU1999 (gagpol), NYU6556-3 (HG1b), BDHS33 (*vif*/*gp120*), NYU124-2 (*gp120*), and MDC131-1 (*gp41*/*nef* and *vifvpu*).

**Figure 4 viruses-11-00317-f004:**
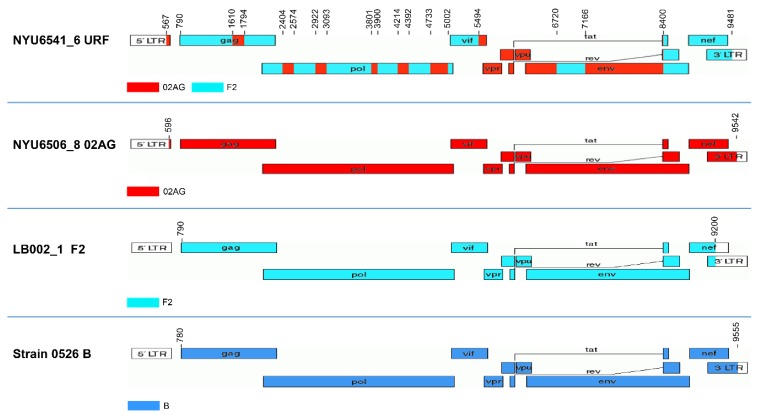
Development of a near full genome sequencing approach using recombinant and pure subtype strains. Genome maps of four HIV-1 strains, which were used to develop the near full genome sequencing approach. The determined subtypes are indicated using the recombinant drawing tool of the LANL database. NYU6541_6, NYU6506_8, LB002_1 and strain 0526 correspond to a unique recombinant form (URF), the circulating recombinant form CRF02_AG, sub-subtype F2, and subtype B, respectively.

**Figure 5 viruses-11-00317-f005:**
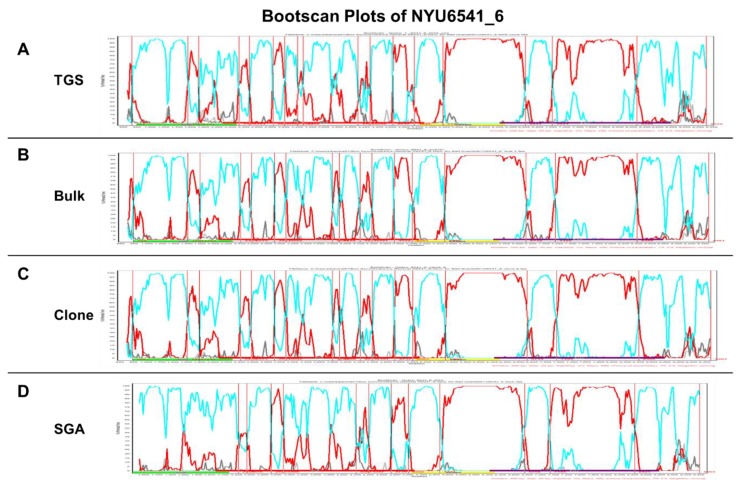
Comparative recombinant breakpoint analysis of near full genome sequences obtained with four different amplification, sequencing, and cloning strategies. Using sample NYU6541_6, different amplification, sequencing, and cloning techniques were studied: (**A**) third-generation sequencing (TGS), (**B**) bulk amplification and Sanger sequencing, (**C**) cloning after bulk amplification, followed by Sanger sequencing, and (**D**) single genome amplification (SGA) and Sanger sequencing. BootScan plots (Simplot software) are shown for near full genome sequences of NYU6541_6 with window size 200 and step size 20. Breakpoint patterns were determined by analyzing the query sequence against CRF02_AG (red), F2 (cyan), C (dark gray), and B (light gray) reference strains. The genomic regions *gag*, *pol*, *vif/vpr/vpu* and *env* are schematically shown along the x-axis in green, red, yellow, and purple bars, respectively. The *y*-axis indicates bootstrap values obtained after 250 replicative measurements.

**Table 1 viruses-11-00317-t001:** Amplification primers.

Portion Analyzed	Nested PCR Round	Primer ID	Sequence	Position in HxB2	Temp.	Reference
Near full genome 1 (NFG1)	1st Round	5’ UTR_For1	TGCCTTGAGTGCTTCAAGTAGTGTG	536–560	58.4 °C	This study
		3’ UTR_Rev1	TATTGAGGCTTAAGCAGTGGGTTC	9615–9592	57 °C	This study
	2nd Round	5’ UTR_For2	TCTGTTGTGTGACTCTGGTAACTAGAGATC	567–596	58.7 °C	This study
		3’ UTR_Rev4	GCTCAAATCTGGTCTAGCAAGAGAGA	9567–9542	58 °C	This study
Near full genome 2 (NFG2)	1st Round	HIV_682_For	TCTCTCGACGCAGGACTCGGCTTGCTG	682–708	67 °C	[25]
		HIV_9555_Rev	TCTACCTAGAGAGACCCAGTACA	9555–9533	55.5 °C	[25]
	2nd Round	HIV_776_For	CTAGAAGGAGAGAGAGATGGGTGCGAG	776–800	61 °C	[25]
		HIV_9555_Rev	TCTACCTAGAGAGACCCAGTACA	9555–9533	55.5 °C	[25]
Near full genome 3 (NFG3)	1st Round	HIV_682_For	TCTCTCGACGCAGGACTCGGCTTGCTG	682–708	67 °C	[25]
		3’ UTR_Rev3	AGAGCTCCCAGGCTCAAATCTGGTCTA	9578–9552	62.5 °C	This study
	2nd Round	HIV_776_For	CTAGAAGGAGAGAGAGATGGGTGCGAG	776–800	61 °C	[25]
		3’ UTR_Rev4	GCTCAAATCTGGTCTAGCAAGAGAGA	9567–9542	58 °C	This study
Near full genome 4 (NFG4)	1st Round	HIV_682_For	TCTCTCGACGCAGGACTCGGCTTGCTG	682–708	67 °C	[25]
		3’ UTR_Rev5	CAGTACAGGCGAGAAGCAGCTGCT	9539–9516	62 °C	This Study
	2nd Round	HIV_776_For	CTAGAAGGAGAGAGAGATGGGTGCGAG	776–800	61 °C	[25]
		3’ UTR_Rev6	AGCAGCTGCTTATATGCAGCATCTGAG	9525–9499	60.7 °C	This Study
Half genome 1a (HG1a)	1st Round	HIV_682_For	TCTCTCGACGCAGGACTCGGCTTGCTG	682–708	67 °C	[25]
		Vpu2_Rev	CCGCTTCTTCCTGCCATAGGA	5985–5966	59.1 °C	This study
	2nd Round	HIV_776_For	CTAGAAGGAGAGAGAGATGGGTGCGAG	776–800	61 °C	[25]
		Vpu3_Rev	TCCTGCCATAGGAGATGCCTAAG	5978–5956	58.1 °C	This study
Half genome 1b (HG1b)	1st Round	Gag3_For	GAGAGATGGGTGCGAGAGC	785–803	57 °C	This study
		Vpu2_Rev	CCGCTTCTTCCTGCCATAGGA	5985–5966	59.1 °C	This study
	2nd Round	Gag4_For	TAGTATGGGCAAGCAGGGA	890–908	56 °C	This study
		Vpu3_Rev	TCCTGCCATAGGAGATGCCTAAG	5978–5956	58.1 °C	This study
Half genome 2 (HG2)	1st Round	Vif2_For	TGGAAAGGTGAAGGGGCAGTA	4956–4976	58.4 °C	This study
		OFM19	GCACTCAAGGCAAGCTTTATTGAGGCTTA	9632–9604	60.9 °C	[31]
	2nd Round	Vif3_For	GATTATGGAAAACAGATGGCAGGT	5037–5060	55.2 °C	This study
		HIV_9555_Rev	TCTACCTAGAGAGACCCAGTACA	9555–9533	55.5 °C	[25]
*gagpol*	1st Round	Gag2_For	GACTAGCGGAGGCTAGAAG	764–782	54 °C	This study
		Pol2_Rev	CCATGTTCTAATCCTCATCCTGTC	5103–5080	55 °C	This study
	2nd Round	Gag3_For	GAGAGATGGGTGCGAGAGC	785–803	57 °C	This study
		Pol3_Rev	CTGTCTACCTGCCACACA	5084–5067	54 °C	This study
*gag*	1st Round	Gag2_For	GACTAGCGGAGGCTAGAAG	764–782	54 °C	This study
		Gag1_Rev	CCAATTCCCCCTATCAT	2404–2388	48 °C	This study
	2nd Round	Gag4_For	TAGTATGGGCAAGCAGGGA	890–908	56 °C	This study
		Gag3_Rev	GGTCGTTGCCAAAGAGTGA	2278–2260	55 °C	This study
*pol*	1st Round	Pol1_For	GAAGAAATGATGACAGC	1819–1835	45 °C	This Study
		Pol1_Rev	TGCCAGTCTCTTTCTCCTG	5279–5161	54 °C	This study
	2nd Round	Pol2_For	AAGTGTTTCAACTGTGG	1960–1976	47 °C	This study
		Pol2_Rev	CCATGTTCTAATCCTCATCCTGTC	5103–5080	55 °C	This study
*vifvpu*	1st Round	Vif 1_For	GGGTTTATTACAGGGACAGCAGAG	4900–4923	57.3 °C	[31]
		Vpu1_Rev	TTGCCACTYTCTTCTGCTCTTTC	6225–6203	56.4 °C	This study
	2nd Round	Vif 2_For	TGGAAAGGTGAAGGGGCAGTA	4956–4976	58.4 °C	This study
		Vpu2_Rev	CCGCTTCTTCCTGCCATAGGA	5985–5966	59.1 °C	This study
*vif*/gp120	1st Round	Vif1_For	GGGTTTATTACAGGGACAGCAGAG	4900–4923	57.2 °C	This study
		Gp120out	GCARCCCCAAAKYCCTAGG	8018–8000	57.2 °C	This study
	2nd Round	Vif2_For	TGGAAAGGTGAAGGGGCAGTA	4956–4976	58.4 °C	This study
		Gp120in	CGTCAGCGTYATTGACGCYGC	7838–7818	61.4 °C	This study
*env*	1st Round	Vif1_For	GGGTTTATTACAGGGACAGCAGAG	4900–4923	57.2 °C	[31]
		OFM19	GCACTCAAGGCAAGCTTTATTGAGGCTTA	9632–9604	60.9 °C	[31]
	2nd Round	Env A	(CACC)GGCTTAGGCATCTCCTATGGCAGGAAGAA	5954–5982	63 °C	[31]
		02AG-Env N	GTTCTGCCAATCTGGGAAGAATCCTTGTGTG	9174–9144	62.3 °C	This study
*gp120*	1st Round	Env A	GGCTTAGGCATCTCCTATGGCAGGAAGAA	5954–5982	62.8 °C	[31]
		Gp120out	GCARCCCCAAAKYCCTAGG	8018–8000	57.2 °C	[39]
	2nd Round	Env B	AGAAAGAGCAGAAGACAGTGGCA	6202–6224	58.2 °C	[31]
		Gp120in	CGTCAGCGTYATTGACGCYGC	7838–7818	61.4 °C	[39]
*gp41*/*nef*	1st Round	Gp120in_For1	CAGCAGGAAGCACTATGGGCG	7798–7818	60.7 °C	This study
		3’UTR_Rev1	TATTGAGGCTTAAGCAGTGGGTTC	9615–9592	57 °C	This study
	2nd Round	Gp120in_For2	GCRGCGTCAATRACGCTGACG	7818–7838	61.4 °C	This study
		3’UTR_Rev4	GCTCAAATCTGGTCTAGCAAGAGAGA	9567–9542	58 °C	This study

**Table 2 viruses-11-00317-t002:** Forward sequencing primers.

Primer ID	Sequence	Position in HxB2	Temp.	Reference
HIV_580_For	TCTGGTAACTAGAGATCC	580–597	46.6 °C	This study
HIV_625_For	ATCTCTAGCAGTGGCGCCCGA	625–641	63 °C	This study
HIV_788_For	AGATGGGTGCGAGAGCGT	788–809	59.2 °C	This study
HIV_1250_For	CATGGGTAAAGGTAATAGAAG	1250–1270	47.9 °C	This study
HIV_1250b_For	CATGGGTAAAAGTAATAGAA	1250–1269	44.5 °C	This study
HIV_1400_For	CCATCAATGAGGAAGCTGCA	1400–1419	55.4 °C	This study
HIV_1830_For	TGATGACAGCATGCCAGG	1830–1847	55.3 °C	This study
HIV_1970_For	TTCAACTGTGGCAAAGAAGG	1970–1989	53.4 °C	This study
HIV_2075_For	GACAGGCTAATTTTTTAGGGA	2075–2095	49.9 °C	This study
HIV_2590_For	CAGGAATGGATGGCCCAA	2590–2607	55.2 °C	This study
HIV_2700_For	GGGCCTGAAAATCCATACAATACT	2700–2723	55.1 °C	This study
HIV_2756_For	GTACTAAATGGAGAAAATTAG	2756–2776	43.8 °C	This study
HIV_3300_For	AGCTGGACTGTCAATGA	3300–3316	50.4 °C	This study
HIV_3350_For	GGGCAAGTCAAATTTATCCAG	3350–3370	51.8 °C	This study
HIV_3355_For	AGCCAGATTTATCCAGG	3355–3371	48.2 °C	This study
HIV_4000_For	TAGCCTTGCAGGATTCAGGAT	4000–4020	56.0 °C	This study
HIV_4175_For	TGGAGGAAATGAACAAGTAGA	4175–4195	51 °C	This study
HIV_4542_For	GCAGGAAGATGGCCAGT	4542–4558	55.0 °C	This study
HIV_4646_For	TGGAATTCCCTACAATCC	4646–4663	48.6 °C	This study
HIV_4747_For	AGACAGCAGTACAGATGGCAG	4747–4767	56.7 °C	This study
HIV_4900_For	GGGTTTATTACAGGGACAGCA	4900–4920	54.6 °C	This study
HIV_5388_For	TTTCAGAATCTGCCATAAG	5388–5406	47.1 °C	This study
HIV_5769_For	CATTTCAGAATYGGGTG	5769–5786	46.5 °C	This study
HIV_5840_For	GTAGATCCTARCCTAGA	5840–5856	44.1 °C	This study
HIV_5970_For	ATGGCAGGAAGAAGCGGAGAC	5970–5990	59.5 °C	This study
HIV_6100_For	AGTAGCATTCATAGCAGCCAT	6100–6120	54.0 °C	This study
HIV_6125_For	GTGTGGACTATAGTATATATAG	6125–6146	44.1 °C	This study
HIV_6380_For	ATTTTGTGCATCAGATGC	6380–6397	44.7 °C	This study
HIV_6543_For	GATATAATTAGTCTATGGG	6543–6561	40.8 °C	This study
HIV_6745_For	CACTTTTTTATAGACTTGAT	6745–6764	42.7 °C	This study
HIV_6826_For	TTAMACAGGCTTGTCC	6826–6841	47.2 °C	This study
HIV_6840_For	CCAAAGGTATCCTTTGAGCCA	6840–6860	54.9 °C	This study
HIV_6855_For	GAGCCAATTCCCATACAT	6855–6872	49.3 °C	This study
HIV_7500_For	ATGTGGCAGAAAGTAGGACAAGC	7500–7522	57.1 °C	This study
HIV_7520_For	AGCAATGTATGCCCCTC	7520–7536	52 °C	This study
HIV_7633_For	CTGGAGGAGGAGATATGAG	7633–7651	50.7 °C	This study
HIV_7660_For	GGAGAAGTGAATTATATAA	7660–7678	40.6 °C	This study
HIV_7803_For	GGAAGCACTATGGGCGC	7803–7819	56.7 °C	This study
HIV_8800_For	GGTGGCAAGTGGTCAAA	8800–8816	53.0 °C	This study
HIV_8180_For	GCAGGAAAAGAATGAACAAG	8180–8199	49.7 °C	This study
HIV_8970_For	GGTTAGAAGCACAAGAG	8970–8986	47.0 °C	This study
HIV_9015_For	GGTACCTTTAAGACCAATGA	9015–9034	49.3 °C	This study
HIV_9030_For	GACCAATGACTTATAAGG	9030–9047	44.1 °C	This study

**Table 3 viruses-11-00317-t003:** Reverse Sequencing Primer.

Primer ID	Sequence	Position in HxB2	Temp.	Reference
HIV_809_Rev	ACGCTCTCGCACCCATCT	809–792	58.9 °C	This Study
HIV_912_Rev	TCCCTGCTTGCCCATACTA	912–894	55.6 °C	This study
HIV_1270_Rev	CTTCTATTACTTTTACCCATG	1270–1250	45.9 °C	This study
HIV_1505_Rev	GTTCCTGCTATRTCACTTCC	1505–1486	51.5 °C	This study
HIV_2095_Rev	TCCCTAAAAAATTAGCCTGTC	2095–2075	50.2 °C	This study
HIV_3577_Rev	TGATAAATTTGATATGTCCA	3577–3558	43.9 °C	This study
HIV_5060_Rev	ACCTGCCATCTGTTTTCCATA	5060–5040	54.2 °C	This study
HIV_5537_Rev	ACACTAGGCAAAGGYGG	5537–5521	53.5 °C	This study
HIV_6352_Rev	GGTACCCCATAATAGACTGTRACCCACAA	6352–6324	59.9 °C	[25]
HIV_6764_Rev	ATCAAGTCTATAAAAAAGTG	6764–6745	42.7 °C	This study
HIV_6841_Rev	GGACAAGCCTGTKTAA	6841–6826	47.2 °C	This study
HIV_6867_Rev	TGGGAATTGGCTCAAA	6867–6852	48.1 °C	This study
HIV_6904_Rev	TTTAGAATCGCAAAACCAGC	6904–6885	51.4 °C	This study
HIV_7025_Rev	TTCTGCTAGRCTGCCATT	7025–7008	52.4 °C	This study
HIV_7085_Rev	CTGTACTATTATGGTTT	7085–7069	39.1 °C	This study
HIV_7351_Rev	AAACTATGTGTTGTAATTTC	7351–7332	43.5 °C	This study
HIV_7819_Rev	GCGCCCATAGTGCTTCC	7819–7803	56.7 °C	This study
HIV_7845_Rev	CCYGTACCGTCAGCGT	7845–7830	56 °C	This study
HIV_8075_Rev	TTTTTACACCGTCTAG	8075–8059	42 °C	This study
HIV_8448_Rev	TGTCTTGCTCKCCACCT	8448–8432	42 °C	This study
HIV_8528_Rev	GTAGCTGAAGAGGCACAG	8528-8511	54.9 °C	This study
HIV_8529_Rev	GGTAGCTGAAGAGGCACAGG	8529-8510	52.6 °C	This study
REV14	ACCATGTTATTTTTCCACATGTTAAA	6526–6501	54.6 °C	[31]
REV15	CTGCCATTTAACAGCAGTTGAGTTGA	7015–6990	57.8 °C	[31]
REV 16	ATGGGAGGGGCATACATTGCT	7540–7520	59 °C	[31]
REV 17	CCTGGAGCTGTTTAATGCCCCAGAC	7956–7932	61.7 °C	[31]
REV 18	GGTGAGTATCCCTGCCTAACTCTAT	8365–8341	57.2 °C	[31]
REV 19	ACTTTTTGACCACTTGCCACCCAT	8820–8797	59.6 °C	[31]

**Table 4 viruses-11-00317-t004:** Summary of amplification approaches and yield using 23 pure or recombinant HIV-1 samples.

		Near Full Genome 1	Near Full Genome 2	Near Full Genome 3	Near Full Genome 4	Half Genome 1a	Half Genome 1b	Half Genome 2	*gagpol*	*vif*/*gp120*	*env*	*gp120*
	1st Rd Primers	5’UTRfor1/ 3’UTRrev1	HIV682for/ HIV9555rev	HIV682for/ 3’UTRrev3	HIV682for/ 3’UTRrev5	HIV682for/ Vpu2 rev	Gag3for/ Vpu2 rev	Vif2for/ OFM19	Gag2for/ pol2 rev	Vif1for/ Gp120out	Vif1/ OFM19	EnvA/ Gp120out
Sample ID	2nd Rd Primers	5’UTRfor2/ 3’UTR rev4	HIV776for/ HIV9555rev	HIV776for/ 3’UTR rev4	HIV776for/ 3’UTR rev6	HIV776for/ Vpu3rev	Gag4for/ Vpu3 rev	Vif3for/ HIV9555rev	Gag3for/ pol3rev	Vif2for/ Gp120in	EnvA/ 02AG-EnvN	EnvB/ Gp120in
LB016-1		positive	−	−	−	positive	positive	positive	−	positive	positive	positive
LB069-1		positive	−	−	−	−	positive	positive	positive	−	positive	positive
LB082-1		−	−	−	−	−	positive	positive	positive	positive	positive	positive
LB089-1		positive	−	positive	−	−	positive	positive	positive	positive	positive	positive
LB095-1		positive	−	−	−	positive	positive	positive	positive	−	positive	positive
LB104-1		−	−	−	−	−	positive	positive	−	positive	−	positive
MDC131-1		−	−	−	−	positive	−	positive	positive	−	−	−
MDC179-2		−	−	−	−	−	positive	positive	positive	positive	−	−
BDHS24-2		−	−	−	−	−	positive	positive	positive	positive	positive	positive
BDHS33		−	−	−	−	−	−	positive	positive	positive	−	positive
NYU119-3		−	−	−	−	positive	positive	positive	−	positive	−	positive
NYU124-2		−	−	−	−	positive	positive	positive	−	positive	positive	positive
NYU129-5		−	positive	−	−	−	positive	positive	positive	−	positive	positive
NYU1122-1		−	positive	−	positive	positive	positive	positive	−	positive	−	positive
NYU1999		−	−	−	−	−	positive	positive	positive	−	−	positive
NYU2140-1		−	−	−	−	positive	−	positive	−	positive	positive	positive
NYU6556-3		−	−	−	positive	−	positive	positive	positive	positive	positive	−
NYU6541-6		positive	positive	positive	positive	positive	positive	positive	−	positive	positive	positive
NYU6506-8		−	−	−	−	−	positive	positive	−	−	−	−
Strain0526		−	−	−	−	positive	−	positive	−	−	−	−
LB002-1		−	−	positive	−	NA	NA	NA	NA	NA	NA	NA
LB022-1		−	−	positive	−	NA	NA	NA	NA	NA	NA	NA
LB006-1		−	−	−	positive	NA	NA	NA	NA	NA	NA	NA
**Miscellaneous HIV-1 genome regions amplified**												
		***gag***	***pol***	***vifvpu***	***gp120***	***gp41/nef***						
	**1st Rd primers**	**Gag2for/** **Gag1 rev**	**Pol1for/** **Pol1rev**	**Vif1for/** **Vpu1rev**	**EnvA/** **Gp120out**	**Gp120in for1/** **3’UTRrev1**						
**Sample ID**	**2nd Rd primers**	**Gag4for/** **Gag3rev**	**Pol2for/** **Pol2rev**	**Vif2for/** **Vpu2rev**	**EnvB/** **Gp120in**	**Gp120in for2/** **3’UTRrev4**						
MDC131-1		positive	positive	positive	NA	positive						
NYU119-3		positive	positive	positive	positive	positive						
NYU124-2		positive	positive	positive	positive	positive						

NA: not applicable; −: negative.

**Table 5 viruses-11-00317-t005:** Summary of third-generation sequencing (TGS) and single-genome amplification (SGA).

#	Sample ID	TGS Total Number of HIV Reads	TGS Average Read Length	TGS Number of Long HIV Reads	TGS Genomic Region	SGA Genomic Region
1	LB016-1	713	1731	17 (>5000 bp)	Near full genome 1	NA
2	LB069-1	1701	3929	488 (>5000 bp)	Near full genome 1	NA
3	LB082-1	610	4232	179 (>4000 bp)	Half genome 2	Half genome 2
4	LB089-1	NA	NA	NA	NA	NA
5	LB095-1	NA	NA	NA	NA	Half genome 2
6	LB104-1	1120	3830	61 (>4000 bp)	Half genome 2	NA
7	MDC131-1	NA	NA	NA	NA	NA
8	MDC179-2	1956	2807	504 (>2000 bp)	*vif*/*gp120*	NA
9	BDHS024-2	3463	4047	799 (>4000 bp)	Half genome 2	NA
10	BDHS33	3184	4122	813 (>4000 bp)	Half genome 2	NA
11	NYU119-3	NA	NA	NA	NA	NA
12	NYU124-2	2076	2837	964 (>2000 bp)	*vif*/*gp120*	NA
13	NYU129-5	NA	NA	NA	NA	Half genome 2
14	NYU1122-1	2648	2478	752 (>2000 bp)	*vif*/*gp120*	NA
15	NYU1999-1	NA	NA	NA	NA	NA
16	NYU2140-1	2108	4044	448 (>4000 bp)	Half genome 2	NA
17	NYU6556-3	NA	NA	NA	NA	Half genome 2
18	NYU6541-6	262	1415	41 (>5000 bp)	Near full genome 1	Near full genome 1
19	NYU6506-8	NA	NA	NA	NA	NA
20	Strain0526	NA	NA	NA	NA	Half genome 2
21	LB002-1	NA	NA	NA	NA	NA
22	LB022-1	NA	NA	NA	NA	NA
23	LB006-1	NA	NA	NA	NA	NA

NA: not applicable.

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
