# Peer review of "Development of a Versatile, Near Full Genome Amplification and Sequencing Approach for a Broad Variety of HIV-1 Group M Variants"

_viruses, 2019, doi:10.3390/v11040317_

Round 1
Reviewer 1 Report
The authors described the optimized HIV-1 near full genome sequencing approach to study the recombinant breakpoints patterns, subtype compositions, particularly for variable subtypes of HIV-1 group M. They newly designed several primers sets for group M from most conservative regions to amplify and sequence the full HIV genome. The authors also compared the different sequencing approaches and performances on dissecting each HIV sample. I have some concerns that I have listed below:
Oxford Nanopore sequencing is not NGS, but third-generation sequencing technology. NGS is specifically for short reads sequencing, such as Illumina. Try to find other abbreviations, if Illumina also works, maybe you can consider using high-throughput sequencing (HTS).
Please explain what is single-genome amplification (SGA).
Please describe clearly what are the DNA input for different sequencing approach in the method section.
Lots of abbreviation and terms are used, some of them are confused or redundant, like HG1 and 1st half genome, HG2 and 2st half genome, NFG, NFGS.
All genes should be italia.
Page 2, 49 - 51: Are subtypes, sub-subtypes, CRFs, and URFs all different things? Maybe more information is needed. Are A1-A5, F1, F2 subtypes or sub-subtypes? It is not consistent with page 7, 217, which are all described as subtypes. A5 should be corrected to A6.
Page 2, 80: cloning is not a sequencing technology, please use the right word.
Page 7, 225: I thought NFG amplicon approach include three different approaches, including two amplicon approach. If so, here NFG should be corrected to One amplicon approach.
Page 7, 250: sequencing reads are confused when NGS is also used in the paper. Maybe you can consider to use fragment instead.
Table 1. Are those primers been used for different approaches, or some primers can be used for multiple approaches? If the first case, maybe you can add the column indicating different approaches, and separating them by row lines which will be clearer.
Move Table 2 and Table 3 to the Supplementary tables.
Table 4. Note to distinguish “NA” and “-”.
Table 5. What is the average length for the “Total Number of HIV reads”. Is Half genome 2 the same as HG2?
Figure 3. The terms used here are also confused, such as NFG, Half genome 1/2, HG 1/2, Partial NFG amplicons? Is HG 1/2 also belong to partial NFG?
Author Response
We thank the Reviewer for the careful editing and for the helpful ideas to improve the manuscript. According to the reviewer’s requests, we have done the following revisions:
(The page/line numbers refer to the revised manuscript in track changes mode.
We added our bioinformatics expert Alireza Khodadadi-Jamayran to the author list, who contributed essentially to the data analysis for MinIon third-generation sequencing.)
Point 1: Oxford Nanopore sequencing is not NGS, but third-generation sequencing technology. NGS is specifically for short reads sequencing, such as Illumina. Try to find other abbreviations, if Illumina also works, maybe you can consider using high-throughput sequencing (HTS).
Response 1: We acknowledged the reviewer's suggestions and changed the term next-generation sequencing (NGS) to third-generation sequencing (TGS) throughout the manuscript. As a general term for NGS and TGS approaches, we used high-throughput sequencing or deep sequencing.
Point 2: Please explain what is single-genome amplification (SGA).
Response 2: A more detailed explanation of SGA is now provided in the Methods section (page 5-6, lines 243-255). In brief, this technique serves to ensure amplification from a single template and elimination of template recombination during PCR.
Point 3: Please describe clearly what is the DNA input for different sequencing approach in the method section.
Response 3: We added the requested information for SGA (page 5, lines 250-255). For bulk sequencing and MinIon third-generation sequencing, this information is already provided.
Point 4: Lots of abbreviation and terms are used, some of them are confused or redundant, like HG1 and 1st half genome, HG2 and 2st half genome, NFG, NFGS.
Response 4: We agree that the terminology is very complex. We simplified the terminology for the first and second half genomes in using the terms “half genome 1 (HG1)” or “half genome 2 (HG2)” consistently throughout the manuscript. We think the abbreviations NFG and NFGS are very useful because their use allows for easier reading of the manuscript. However, we are open to further suggestions to improve the terminology and understanding of the manuscript.
Point 5: All genes should be italia.
Response 5: We thank the reviewer for careful editing. We italicized several gene names throughout the manuscript as can be seen in the track change mode of the manuscript. Compound terms built of English words such as “near full genome” or “half genome 1” were left in roman type (not italicized). However, if the reviewer and the editor would prefer to have this changed, we will be glad to oblige.
Point 6: Page 2, 49 - 51: Are subtypes, sub-subtypes, CRFs, and URFs all different things? Maybe more information is needed. Are A1-A5, F1, F2 subtypes or sub-subtypes? It is not consistent with page 7, 217, which are all described as subtypes. A5 should be corrected to A6.
Response 6: Yes, they are all different: sub-subtypes are sub-categories of subtypes. CRFs and URFs are recombinant forms composed of at least two (sub)-subtypes. A CRF is defined as a recombinant form (former URF), identified in at least three epidemiologically unlinked individuals. According to the reviewer’s suggestion, we added more information to the introduction (page 2, lines 60-62), we changed the term subtypes into (sub)-subtypes in the results (page 9, line 313), and we corrected A5 to A6 in the introduction (page 2, line 58).
Point 7: Page 2, 80: cloning is not a sequencing technology, please use the right word.
Response 7: We could not find a wrong wording on page 2, line 80 (please advise if we overlooked), but we improved the wording with regards to cloning on page 15, line 427 and in Figure legend 5, on page 16, lines 442-443.
Point 8: Page 7, 225: I thought NFG amplicon approach include three different approaches, including two amplicon approach. If so, here NFG should be corrected to One amplicon approach.
Response 8: We are not sure to which passage of the text the reviewer refers since line 225 does not apply. In case the reviewer referred to line 254 (initial version of the manuscript), the wording seems right to us because it describes amplification (and not sequencing). Distinguishing NFG amplification and NFG sequencing is essential in this regard. NFG amplification always describes the one amplicon approach. With regards to NFG sequencing, different amplicon strategies exist. It is important to distinguish whether the NFG sequence is composed of sequences from one, two, or multiple amplicons.
Point 9: Page 7, 250: sequencing reads are confused when NGS is also used in the paper. Maybe you can consider to use fragment instead.
Response 9: As suggested by the reviewer, the terms were changed in the text (page 11, lines 354-355).
Point 10: Table 1. Are those primers been used for different approaches, or some primers can be used for multiple approaches? If the first case, maybe you can add the column indicating different approaches, and separating them by row lines which will be clearer.
Response 10: Some primers of Table 1 were used for the amplification of different (=multiple) constructs, for example, OFM19 was used for the amplification of HG2 and env, or 3’UTR Rev4 was used for NFG1, NFG3, and gp41/nef. Therefore, the primers could not be subclassified according to the amplified region. We hope our answer addressed the reviewer’s question.
Point 11: Move Table 2 and Table 3 to the Supplementary tables.
Response 11: The authors of the paper think that the design and introduction of a novel set of universal primers for the amplification and sequencing of diverse HIV-1 group M (sub)-subtypes and recombinant forms is a significant strength of the paper and might be useful for a broader audience. We agree with the reviewer that the amplification primers are of primary importance, but also the sequencing primers have proven very important. Therefore, we would suggest leaving Tables 2 and 3 in the main part of the manuscript; however, if the reviewer and editor are convinced that they are better suited for the Supplementary material, we will gladly make these changes.
Point 12: Table 4. Note to distinguish “NA” and “-”.
Response 12: We thank the reviewer for the thorough editing. Explanations for “NA” and “-“ are added as a footnote to the table.
Point 13: Table 5. What is the average length for the “Total Number of HIV reads”. Is Half genome 2 the same as HG2?
Response 13: As requested, we have added a column to Table 5 showing the average read lengths. Half genome 2 is the same as HG2 as we defined on page 8, line 296.
Point 14: Figure 3. The terms used here are also confused, such as NFG, Half genome 1/2, HG 1/2, Partial NFG amplicons? Is HG 1/2 also belong to partial NFG?
Response 14: According to the reviewer’s request, we simplified the terminology in Figure 3. Half genome 1/2 (HG1/2) is consistently used and the abbreviations are defined both in the results text and in the Figure legend. To avoid confusion, we removed the term “partial NFG amplicons” and replaced “partial NFG amplicons within the 1st / 2nd half genome” with “diverse amplicons within half genome 1/2 (HG1/2)”.

Reviewer 2 Report
Summary:
This paper describes a method for near-full-genome amplification of M group HIV-1 variants which can be applied to a wide variety of subtypes and circulating recombinant forms. The main contribution of this study is the primer design and flexible amplification strategy utilizing single, double, or multiple-amplicons for RT-PCR amplification of HIV-1 RNA in plasma samples which can be applied to a wide variety of HIV-1 subtypes. Multiple near full genome sequencing strategies for HIV-1 have already been published. The novel contribution of this study is the application to a variety of viral subtypes and recombinant forms of HIV-1 and the explicit consideration of making the amplification strategy suitable for a range of downstream experimental applications, i.e. next-generation sequencing, sanger sequencing, single genome amplification, and molecular cloning.
In general, the strengths of the manuscript are that (1) the manuscript is overall well-organized, and the literature is well-synthesized providing a clear framework for the significance of the current study; (2) the amplification strategy described is capable of amplifying viral genomes from more M group subtypes than existing published methods; (3) the amplification strategy using variable numbers of genome-spanning amplicons is more robust than a strategy using a set number of amplicons; (4) although the novel contributions of this study are incremental, if the approach does prove robust on a larger sample set (>23) then the potential impact on current knowledge will be greater than the specific results presented in this paper; (5) the experimental design is sound, the validation of the approach was thorough, and the Results are in keeping with state-of-the-art data quality; (6) the techniques are thoroughly described and should be able to be implemented in other laboratories.
The major weaknesses are as follows:
1. While the downstream impact of implementing the current approach is potentially significant, the novelty of the study is subtle. The techniques utilized in this study are all well-established and heavily relied upon in the field. The novelty of the current approach lies largely in the choice of primers and the identification of conserved genomic regions in which to place them. Rational primer design is referenced throughout the manuscript, but the process is not described. While it is clear that the authors are not intending to introduce a pipeline for primer design but rather a comprehensive set of primers and protocols, a discussion of how the primers were designed, screened, and selected seems relevant. Especially because the primer choices represent a key component of the novelty of this study.
2. The relatively small sample size in the study is a weakness in the validation of the authors’ approach. The 23 samples used in the study represent a range of subtypes which is compelling in the argument that this approach is broadly/universally applicable. But even within the small sample set, a varying-number-of-amplicons strategy is necessary for near full genome amplification in all samples. This manuscript is not introducing a pipeline for designing multiple-amplicon strategies but rather a set of primers and protocols to be broadly applied as an NFGS strategy. With this in mind a more thorough discussion of why the authors believe that the presented primers and strategy will be robust on a larger sample set in any or all of these subtypes would seem warranted. This may tie in naturally with the response to the previous comment.
3. Next generation sequencing validation was performed on a MinIon device. Next generation sequencing on other devices such as an Illumina Nextera is more sensitive and capable of identifying minority variants in quasispecies populations. Additional comments on why the authors expect their method to capture minority variants at rates detectable by state-of-the-art NGS systems would be helpful to the discussion as the methods described here remain un-tested for this important downstream application.
4. Figure 3 should mention what samples/subtypes were used to amplify the fragments shown on the gels.
5. The significance of the bootscan plots in Figure 5 could be better explained in the text.
Author Response
The authors acknowledge the sophisticated comments made by the reviewer, which helped to improve the manuscript. We have worked on addressing all points.
(The page/line numbers refer to the revised manuscript in track changes mode.
We added our bioinformatics expert Alireza Khodadadi-Jamayran to the author list, who contributed essentially to the data analysis for MinIon third-generation sequencing.)
Point 1: While the downstream impact of implementing the current approach is potentially significant, the novelty of the study is subtle. The techniques utilized in this study are all well-established and heavily relied upon in the field. The novelty of the current approach lies largely in the choice of primers and the identification of conserved genomic regions in which to place them. Rational primer design is referenced throughout the manuscript, but the process is not described. While it is clear that the authors are not intending to introduce a pipeline for primer design but rather a comprehensive set of primers and protocols, a discussion of how the primers were designed, screened, and selected seems relevant. Especially because the primer choices represent a key component of the novelty of this study.
Response 1: We acknowledged the reviewer's suggestions and, in addition to the short section in the results about rational primer design, we now included a more detailed description on how we designed, screened, and selected the introduced primers (see Methods, page 3-4, lines 135-173).
Point 2: The relatively small sample size in the study is a weakness in the validation of the authors’ approach. The 23 samples used in the study represent a range of subtypes which is compelling in the argument that this approach is broadly/universally applicable. However, even within the small sample set, a varying-number-of-amplicons strategy is necessary for near full genome amplification in all samples. This manuscript is not introducing a pipeline for designing multiple-amplicon strategies but rather a set of primers and protocols to be broadly applied as an NFGS strategy. With this in mind, a more thorough discussion of why the authors believe that the presented primers and strategy will be robust on a larger sample set in any or all of these subtypes would seem warranted. This may tie in naturally with the response to the previous comment.
Response 2: We thank the reviewer for bringing up this important point. We included a more thorough discussion of why we believe the introduced primers and NFGS strategy are suited for a broader use (Discussion, page 20, lines 512-520).
Point 3: Next generation sequencing validation was performed on a MinIon device. Next generation sequencing on other devices such as an Illumina Nextera is more sensitive and capable of identifying minority variants in quasispecies populations. Additional comments on why the authors expect their method to capture minority variants at rates detectable by state-of-the-art NGS systems would be helpful to the discussion as the methods described here remain un-tested for this important downstream application.
Response 3: The authors agree with the reviewer that the sensitive detection of quasispecies using state-of-the-art NGS platforms is of central importance to the field. We included additional information on how our approach can fit in with such downstream procedures (Discussion, page 20, lines 536-539).
Point 4: Figure 3 should mention what samples/subtypes were used to amplify the fragments shown on the gels.
Response 4: We thank the reviewer for the careful editing of the manuscript. We added the requested information to the figure legend.
Point 5: The significance of the bootscan plots in Figure 5 could be better explained in the text.
Response 5: As requested by the reviewer, we gave additional explanations on the significance of the bootscan plots in Figure 5 in the respective section of Results (page 19, lines 468-473). The main significance of the comparative bootscan plots was to compare the NFGS results across platforms in terms of accurate subtype/recombinant form determination.